

# Risk factors and prediction models for recurrent acute ischemic stroke: a retrospective analysis

Liuhua Ke[1], Hongyu Zhang[1], Kang Long[1], Zheng Peng[1],
Yongjun Huang[2], Xingxuan Ma[1] and Wanjun Wu[1]

[1] Department of Clinical Laboratory, Liuzhou Traditional Chinese Medical Hospital, Liuzhou,
Guangxi, China
[2] Department of Neurology, Liuzhou Traditional Chinese Medical Hospital, Liuzhou, Guangxi,
China

## ABSTRACT

**Background:** Ischemic stroke is one of the leading causes of disability and death worldwide, with a high risk of recurrence that severely impacts the quality of life of patients. Therefore, identifying and analyzing the risk factors for recurrent ischemic stroke is crucial for the prevention and management of this disease.

**Methods:** A total of 114 cases of recurrent acute ischemic stroke patients admitted from July 2017 to March 2021 were selected as the observation group, and another 409 cases of initial ischemic stroke patients from the same period as the control group. The clinical data of the observation group and the control group were compared to analyze the risk factors associated with the readmission of ischemic stroke. A single-factor analysis (Model 1), Least Absolute Shrinkage and Selection Operator (LASSO) regression, and machine learning methods (Model 2) were used to screen important variables, and a multi-factor COX Proportional Hazards Model regression stroke recurrence risk prediction model was constructed. The predictive performance of the model was evaluated by the consistency index (C-index).

**Results:** Multivariate COX regression analysis revealed that history of hypertension (Hazard Ratio [HR] = 2.549; 95% Confidence Interval (CI) [1.503–4.321]; $P = 0.001$), history of cerebral infarction (HR = 1.709; 95% CI [1.066–2.738]; $P = 0.026$), cerebral artery stenosis (HR = 0.534; 95% CI [0.306–0.931]; $P = 0.027$), carotid arteriosclerosis (HR = 1.823; 95% CI [1.137–2.924]; $P = 0.013$), systolic blood pressure (HR = 0.981; 95% CI [0.971–0.991]; $P < 0.0001$), red cell distribution width-coefficient of variation (RDW-CV) (HR = 1.251; 95% CI [1.019–1.536]; $P = 0.033$), mean platelet volume (MPV) (HR = 1.506; 95% CI [1.148–1.976]; $P = 0.003$), uric acid (UA) (HR = 0.995; 95% CI [0.991–1.000]; $P = 0.049$) were found significantly associated with acute ischemic stroke. The C-index of the full COX model was 0.777 (0.732~0.821), showing a good discrimination between Model 1 and Model 2.

**Conclusions:** History of hypertension, history of cerebral infarction, cerebral artery stenosis, carotid atherosclerosis, systolic blood pressure, UA, RDW-CV, and MPV were identified as risk factors for acute ischemic stroke recurrence. The model can be used to predict the recurrence of acute ischemic stroke.

Corresponding author
Liuhua Ke, 875874932@qq.com

# INTRODUCTION

Stroke poses a significant global health concern. Globally, approximately one-sixth of individuals will suffer from a stroke during their lifetime (*Figueroa et al., 2021*). According to World Health Organization data, stroke annually causes nearly six million deaths, making it the second leading cause of death worldwide (*Ding et al., 2024*; *Katan & Luft, 2018*; *Tan & Venketasubramanian, 2022*). In addition to primary prevention, several medical organizations and research institutions emphasize the importance of secondary prevention of stroke recurrence. Studies indicate that the recurrence rate of stroke is 7.7% at 3 months, 9.5% at 6 months, and 10.4% at 1 year (*Lin et al., 2021*). Compared to primary strokes, recurrent strokes result in repeated hospital admissions, which often require more complex treatment and more intensive post-death care. This repeated hospitalization not only exacerbates the negative impact on patients' functional status, psychological and emotional well-being, but also significantly increases the financial burden (*Arsava et al., 2016*). In addition, the mortality rate after recurrent ischemic stroke is significantly higher (*Khanevski et al., 2019*; *Liu et al., 2024*; *Skajaa et al., 2022*), further highlighting the importance of preventing recurrent stroke. Given that China is one of the countries with the highest incidence rates of stroke globally, with ischemic stroke making up about 70% of all stroke cases, and considering the high rate of recurrence, it is crucial to identify the risk factors for recurrent ischemic stroke and to establish a predictive model for such recurrences.

In recent years, the prediction model for stroke recurrence has attracted increasing interest (*Akamatsu, Chaitin & Hanafy, 2022*; *Kolmos, Christoffersen & Kruuse, 2021*; *Kozyolkin, Kuznietsov & Novikova, 2019*; *Vodencarevic et al., 2022*; *Wang et al., 2023*), particularly in regions with high incidence. Machine learning approaches have demonstrated significant advantages over conventional statistical methods, particularly in their ability to process high-dimensional data and complex interactions between variables (*Jiang et al., 2024*). For example, Least Absolute Shrinkage and Selection Operator (LASSO) regression can reduce some coefficients to zero and effectively identify key variables related to stroke recurrence, thereby streamlining variable selection, simplifying the model, reducing the risk of overfitting (*Wang et al., 2024*), and improving the model's generalizability, interpretability, and predictive performance. Random forests improve the overall model performance of the model by constructing multiple decision trees and synthesizing their predictive results (*Lee et al., 2024*). This method has strong resistance to overfitting and is less affected by multicollinearity among variables, further enhancing the robustness and accuracy of the model.

This study aims to construct a machine learning model that leverages electronic health record data to predict the risk of recurrence in patients with acute ischemic stroke (AIS) for up to 3 years. Through bootstrap sampling, LASSO regression, and random forest algorithms, the model integrates clinical and laboratory information to provide an

innovative approach for predicting the recurrence risk in AIS patients. Our work not only opens new avenues for personalized treatment strategies for AIS patients but also establishes a robust theoretical framework for future clinical practice.

## MATERIALS AND METHODS

### Data sources, inclusion criteria, and exclusion criteria

The data used in this study were sourced from patients with AIS admitted to the emergency green channel of the Liuzhou Traditional Chinese Medicine Hospital from July 2017 to March 2021. The inclusion criteria for AIS patients were as follows: (1) admitted within 24 h of onset; (2) meeting the diagnostic criteria for AIS *via* imaging examinations. The exclusion criteria included: (1) acute myocardial infarction during hospitalization; (2) severe liver and kidney dysfunction; (3) malignant tumors; (4) death upon discharge; (5) incomplete patient data. All participants were informed of the study and signed consent documents, and the study was ratified by the Medical Ethics Committee of the hospital (2021MAY-KY-YN-009-01).

### Data extraction

Through the stroke green channel timing control system and electronic medical record system of Liuzhou Traditional Chinese Medicine Hospital, data on the first admission was extracted from each patient, including demographic information (gender, age); vital signs (admission systolic blood pressure (SBP), admission diastolic blood pressure (DBP)); vascular risk factors (smoking, drinking); medical history (hypertension, diabetes, coronary heart disease, atrial fibrillation, cerebral infarction, cerebral hemorrhage, cerebrovascular history (stenosis, occlusion, sclerosis), carotid history (stenosis, sclerosis, occlusion, plaque)); assessments based on the National Institutes of Health Stroke Scale (NIHSS) score upon admission; and laboratory tests (white blood cell count (WBC), lymphocyte percentage (LYMPH%), neutrophil percentage (NEUT%), monocyte percentage (MONO%), eosinophil percentage (EO%), basophil percentage (BASO%), red blood cell count (RBC), hemoglobin (HGB), hematocrit (HCT), mean corpuscular volume (MCV), mean corpuscular hemoglobin content (MCH), mean corpuscular hemoglobin concentration (MCHC), red blood cell distribution width-coefficient of variability (RDW-CV), RDW standard deviation (RDW-SD), platelet count (PLT), platelet distribution width (PDW), mean platelet volume (MPV), platelet large cell ratio (P-LCR), plateletcrit (PCT), ultrasensitive C-reactive protein (CRP), prothrombin time (PT), fibrinogen (Fg), thrombin time (TT), activated partial thromboplastin time (APTT), D-Dimer, creatinine (CREA), urea (BUN), uric acid (UA), cystatin C (CYC), blood $\beta$2-microglobulin (MG), alanine aminotransferase (ALT), aspartate aminotransferase (AST), creatine kinase (CK), random glucose (GLU), total cholesterol (TC), triglycerides (TG), high-density lipoprotein cholesterol (HDL-C), low-density lipoprotein cholesterol (LDL-C), homocysteine (HCY), and glycated hemoglobin (HbA1c). Cardiovascular items contained neutrophils/lymphocytes (NLR), platelets/lymphocytes (PLR), monocytes/HDL (MHR), UA/HDL, TG/CYC, CH/HDL, LDL/HDL, and TG/HDL.

### Follow-up

In this study, AIS in patients without a history of stroke was defined as the first stroke event; otherwise, it was considered a recurrent event. Patients were followed up from the first admission to death or discharge. When the surviving patients were discharged, their survival status and recurrent strokes were tracked until March 31, 2023. The primary outcome was the recurrence of AIS. Diagnostic criteria for AIS recurrence were as follows: new signs and symptoms of neurological deficit based on stable or improved symptoms and signs of initial AIS, and a new ischemic lesion confirmed by imaging tests such as cranial CT or magnetic resonance imaging.

### Statistical analyses

Statistical analyses were implemented using R 4.3.1 software (*R Core Team, 2023*), and *P* < 0.05 implied marked differences. Measurement data that conformed to normal distribution were depicted as $\bar{\chi} \pm s$ and analyzed using a *t*-test for intergroup comparisons, and those that were not normally distributed were displayed as M(IQR) and processed by Mann-Whitney U test for intergroup comparisons. Pairwise comparisons of count data were done using the chi-square test. When the percentage of missing variables reached 80% or above, the variables were directly deleted; when between 20% and 80%, continuous variables were converted into count data according to the normal range of the test value, and categorical variables were directly converted into count data with the addition of the "missing" category; when <20%, the missing data were filled in by random forest interpolation. Two models were constructed using univariate regression, LASSO regression, and machine learning: (Model 1) variables identified in univariate analysis were included in the multivariate COX regression; (Model 2) 100 new datasets were randomly generated through the bootstrap method with replacement, and the cv.glmnet model in R was used to select the most important predictive variables for each dataset through 10-fold cross-validation. The top 33 variables that appeared most frequently in the 100 modeling iterations were selected as key variables by LASSO regression. Then, the rfsrc function in R was used to model the original dataset with the random forest method, with the top 35 important variables identified as key variables by machine learning methods. After combining the key variables identified by both methods, they were included in a multivariate COX regression model. The consistency index (C-index) and its 95% confidence interval were calculated to assess the predictive performance of the models. A simplified multifactor COX regression model was built with the important variables screened out by the preferred model, and a risk nomogram was plotted to show the predictive details of the model.

## RESULTS

### Comparison of clinical data

This study included 828 patients with AIS. According to the inclusion and exclusion criteria, 687 patients were diagnosed with AIS and included. Through further screening, 523 eligible patients were included in the study. The patient selection process is detailed in

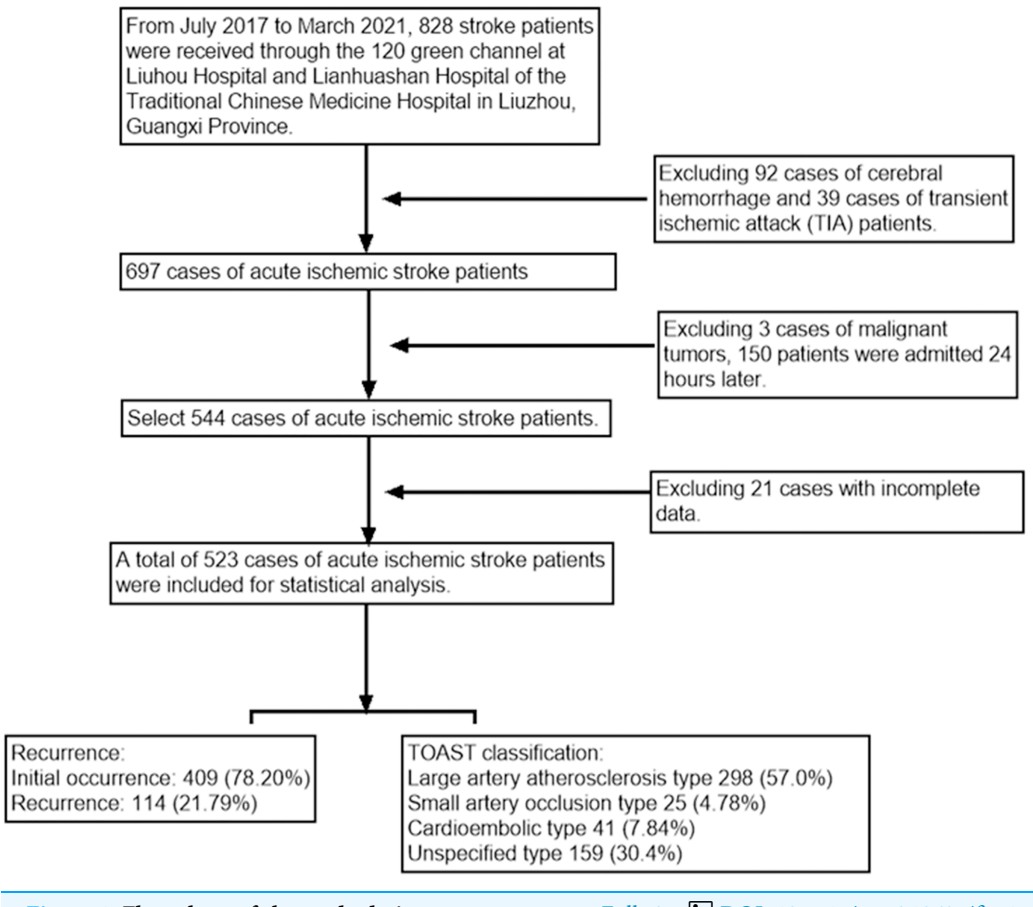

**Figure 1 Flow chart of the study design.**

Fig. 1. In this study, the median (interquartile range) age of AIS patients was 68.00 [59.00, 76.00], with males accounting for 64.4%. According to the diagnostic criteria, 409 non-recurrent patients were assigned to the control group, while 114 recurrent patients (with a recurrence rate of 21.80%) were assigned to the observation group.

In the observation group, older male patients with a history of hypertension and the absence of cerebral artery occlusion were at an increased risk for recurrent large artery atherosclerotic AIS. The increased risk of recurrent stroke may be attributed to vascular wall damage and atherosclerosis resulting from long-term hypertension. In contrast, patients with cerebral artery occlusion may have received treatments for atherosclerosis, such as statins or antiplatelet medications, which may reduce the risk of recurrent stroke.

After data comparison, we found that factors such as TOAST classification, history of hypertension, history of cerebral infarction, cerebral artery occlusion, age, SBP, DBP, NIHSS score, WBC, and GLU were all associated with poor prognosis. In particular, GLU, SBP, and DBP levels of first-time AIS patients were higher than those of recurrent AIS patients, which may indicate that recurrent patients experienced better control of their GLU and blood pressure, effective vascular protective measures, and more aggressive treatment strategies (Table 1).

**Table 1 Clinical characteristics.**

| Items | Total cases N = 523 | Control n = 409 | Observation n = 114 | P |
|---|---|---|---|---|
| Gender, n (%) | | | | 0.513 |
| Male | 337 (64.4%) | 267 (65.3%) | 70 (61.4%) | |
| Female | 186 (35.6%) | 142 (34.7%) | 44 (38.6%) | |
| TOAST type | | | | 0.003 |
| Large-artery atherosclerosis, n (%) | 298 (57.0%) | 243 (59.4%) | 55 (48.2%) | |
| Small artery occlusion, n (%) | 25 (4.78%) | 24 (5.87%) | 1 (0.88%) | |
| Cardiogenic infarction, n (%) | 41 (7.84%) | 32 (7.82%) | 9 (7.89%) | |
| Unknown cause, n (%) | 159 (30.4%) | 110 (26.9%) | 49 (43.0%) | |
| History of hypertension, n (%) | | | | 0.004 |
| No | 200 (38.2%) | 170 (41.6%) | 30 (26.3%) | |
| Yes | 323 (61.8%) | 239 (58.4%) | 84 (73.7%) | |
| History of diabetes, n (%) | | | | 0.65 |
| No | 414 (79.2) | 326 (79.7) | 88 (77.2) | |
| Yes | 109 (20.8) | 83 (20.3) | 26 (22.8) | |
| History of coronary heart disease, n (%) | | | | 0.086 |
| No | 438 (83.7) | 349 (85.3) | 89 (78.1) | |
| Yes | 85 (16.3) | 60 (14.7) | 25 (21.9) | |
| History of atrial fibrillation, n (%) | | | | 0.118 |
| No | 472 (90.2) | 374 (91.4) | 98 (86.0) | |
| Yes | 51 (9.8) | 35 (8.6) | 16 (14.0) | |
| History of CI, n (%) | | | | 0.008 |
| No | 375 (71.7%) | 305 (74.6%) | 70 (61.4%) | |
| Yes | 148 (28.3%) | 104 (25.4%) | 44 (38.6%) | |
| History of cerebral hemorrhage, n (%) | | | | 0.994 |
| No | 507 (96.9) | 397 (97.1) | 110 (96.5) | |
| Yes | 16 (3.1) | 12 (2.9) | 4 (3.5) | |
| Cerebral artery stenosis, n (%) | | | | 0.493 |
| No | 144 (27.5) | 116 (28.4) | 28 (24.6) | |
| Yes | 379 (72.5) | 293 (71.6) | 86 (75.4) | |
| Cerebral artery occlusion, n (%) | | | | 0.003 |
| No | 449 (85.9) | 341 (83.4) | 108 (94.7) | |
| Yes | 74 (14.1) | 68 (16.6) | 6 (5.3) | |
| Carotid stenosis, n (%) | | | | 0.824 |
| No | 482 (92.2%) | 378 (92.4%) | 104 (91.2%) | |
| Yes | 41 (7.84%) | 31 (7.58%) | 10 (8.77%) | |
| Carotid atherosclerosis, n (%) | | | | 0.355 |
| No | 338 (64.6) | 269 (65.8) | 69 (60.5) | |
| Yes | 185 (35.4) | 140 (34.2) | 45 (39.5) | |
| Age (year) | 68.00 [59.00, 76.00] | 67.00 [59.00, 76.00] | 69.00 [62.00, 79.00] | 0.041 |
| SBP (mmHg) | 157.00 [138.00, 176.00] | 160.00 [139.00, 178.00] | 149.50 [136.00, 162.50] | 0.001 |
| DBP (mmHg) | 89.00 [78.00, 100.00] | 89.00 [79.00, 101.00] | 86.00 [76.00, 95.00] | 0.012 |

| Table 1 (continued) | | | | |
|---|---|---|---|---|
| Items | Total cases N = 523 | Control n = 409 | Observation n = 114 | P |
| NIHSS score | 6.00 [3.00, 11.00] | 6.00 [3.00, 12.00] | 5.00 [2.00, 7.85] | 0.003 |
| WBC (×109/L) | 8.40 [6.80, 10.22] | 8.46 [6.90, 10.48] | 7.72 [6.64, 9.64] | 0.021 |
| RBC (×1,012/L) | 4.67 [4.27, 5.05] | 4.67 [4.28, 5.06] | 4.66 [4.22, 4.98] | 0.269 |
| HGB (g/L) | 139.00 [126.00, 149.00] | 140.00 [126.00, 150.00] | 136.00 [124.25, 147.00] | 0.165 |
| RDW-CV (%) | 12.90 [12.30, 13.60] | 12.90 [12.30, 13.60] | 12.85 [12.40, 13.60] | 0.752 |
| PLT (×109/L) | 236.00 [200.50, 287.00] | 237.00 [201.00, 290.00] | 235.00 [193.50, 274.00] | 0.386 |
| MPV (fL) | 8.90 [8.40, 9.60] | 8.90 [8.30, 9.60] | 9.00 [8.50, 9.60] | 0.325 |
| CRP (mg/L) | 3.39 [1.33, 8.50] | 3.16 [1.32, 7.29] | 4.33 [1.36, 9.35] | 0.298 |
| D-Dimer (ug/mL) | 0.83 [0.36, 1.56] | 0.80 [0.35, 1.61] | 1.02 [0.40, 1.48] | 0.565 |
| CREA (umol/L) | 75.50 [61.10, 91.75] | 75.30 [61.40, 91.80] | 75.80 [60.42, 90.85] | 0.699 |
| BUN (mmol/L) | 5.41 [4.30, 6.70] | 5.44 [4.28, 6.70] | 5.35 [4.45, 6.97] | 0.823 |
| UA (umol/L) | 381.00 [316.00, 447.50] | 384.00 [320.00, 452.00] | 368.50 [301.50, 429.25] | 0.218 |
| CYC (mg/L) | 1.13 [0.92, 1.45] | 1.12 [0.91, 1.44] | 1.19 [0.93, 1.48] | 0.508 |
| GLU (mmol/L) | 7.08 [5.81, 8.80] | 7.22 [5.90, 8.87] | 6.65 [5.46, 8.10] | 0.016 |
| HbA1c (%) | 6.20 [5.50, 6.71] | 6.30 [5.50, 6.80] | 6.20 [5.53, 6.60] | 0.366 |
| TC (mmol/L) | 4.73 [4.04, 5.37] | 4.76 [4.06, 5.43] | 4.65 [3.87, 5.24] | 0.15 |
| TG (mmol/L) | 1.30 [0.92, 1.74] | 1.35 [0.92, 1.75] | 1.23 [0.88, 1.71] | 0.266 |
| LDL-C (mmol/L) | 3.09 [2.41, 3.61] | 3.10 [2.45, 3.62] | 3.06 [2.34, 3.56] | 0.225 |
| HDL-C (mmol/L) | 1.10 [0.93, 1.27] | 1.10 [0.92, 1.28] | 1.10 [0.94, 1.25] | 0.922 |

## Performance comparisons of Model 1 and Model 2

### Model 1

A total of 26 variables with $P < 0.2$ in univariate regression were included in a multivariate COX model (Model 1). The results showed that five variables, including a history of hypertension (HR = 1.918, $P = 0.008$), history of CI (HR = 1.718, $P = 0.010$), SBP (HR = 0.986, $P = 0.005$), RBC (HR = 0.572, $P = 0.043$), and RDW-CV (HR = 1.245, $P = 0.016$) were substantially associated with AIS recurrence ($P < 0.05$).

### Model 2

A total of 100 completely new datasets were generated using the Bootstrap method, and LASSO regression was applied for variable screening to extract the top 33 variables with the highest frequency in these 100 model trainings. These variables were included in a multivariate COX regression model along with the top 35 important variables screened in the random survival forest to establish a model for stroke recurrence risk (Model 2). The results discovered that history of hypertension (HR = 2.549, $P = 0.001$), history of CI (HR = 1.709, $P = 0.026$), cerebral arterial stenosis (HR = 0.534, $P = 0.027$), carotid atherosclerosis (CAS) (HR = 1.823, $P = 0.013$), SBP (HR = 0.981, $P < 0.0001$), RDW-CV (HR = 1.251, $P = 0.033$), MPV (HR = 1.506, $P = 0.003$), and UA (HR = 0.995, $P = 0.049$) were markedly linked to AIS recurrence ($P < 0.05$).

**Table 2 Performance comparison of Model 1 and Model 2.**

| Variables | Model 1 (univariate COX model) | | | Model 2 (LASSO and machine learning) | | |
|---|---|---|---|---|---|---|
| | HR | CI | P | HR | CI | P |
| History of hypertension | 1.918 | [1.187–3.100] | 0.008 | 2.549 | [1.503–4.321] | 0.001 |
| History of CI | 1.718 | [1.137–2.596] | 0.010 | 1.709 | [1.066–2.738] | 0.026 |
| SBP | 0.986 | [0.976–0.996] | 0.005 | 0.981 | [0.971–0.991] | <0.001 |
| RBC | 0.572 | [0.333–0.981] | 0.043 | — | — | — |
| RDW-CV | 1.245 | [1.042–1.488] | 0.016 | 1.251 | [1.019–1.536] | 0.033 |
| Cerebral artery stenosis | — | — | — | 0.534 | [0.306–0.931] | 0.027 |
| CAS | — | — | — | 1.823 | [1.137–2.924] | 0.013 |
| MPV | — | — | — | 1.506 | [1.148–1.976] | 0.003 |
| UA | — | — | — | 0.995 | [0.991–1.000] | 0.049 |

*Comparison*

The C-index of Model 1 and Model 2 was 0.719 (0.677–0.761) and 0.777 (0.732–0.821), respectively, and the C-index of Model 2 was higher than that of Model 1, suggesting that Model 2 had better predictive power. Overlapping the variables screened in Model 1 and Model 2 showed that history of hypertension, history of CI, SBP, and RDW-CV were greatly correlated with the risk of stroke recurrence (Table 2), suggesting the importance of these variables in measuring and predicting the risk of stroke recurrence. These findings have important implications for further research and clinical practice.

## Nomogram for risk of AIS recurrence

In Model 2, we screened out eight key risk factors: history of hypertension, history of CI, cerebral artery stenosis, CAS, SBP, UA, RDW-CV, and MPV. To visualize the impact of these risk factors on the prediction of the risk of AIS recurrence, nomograms were plotted based on these factors (Fig. 2).

## DISCUSSION

AIS is a serious health challenge that can lead to long-term disability or even death. For these patients, readmission to the hospital due to recurrence is a common but grave problem that may signal worsening conditions or chronic complications, adversely impacting the patient's quality of life and prognosis and increasing the stress and burden on the healthcare system. Therefore, an in-depth understanding of the factors affecting the recurrence of AIS patients and predicting their recurrence risk is of paramount importance in improving clinical management outcomes and prognosis.

In the present study, we employed an innovative statistical approach to construct a risk prediction model for the recurrence of AIS, surpassing traditional univariate or bivariate models in multiple aspects (Li et al., 2021; Wang et al., 2022; Wu et al., 2023). Utilizing bootstrap resampling, we generated 100 independent datasets and identified the top 33 variables with the highest frequency of occurrence in model training through LASSO regression. These variables were then integrated with the top 35 significant variables

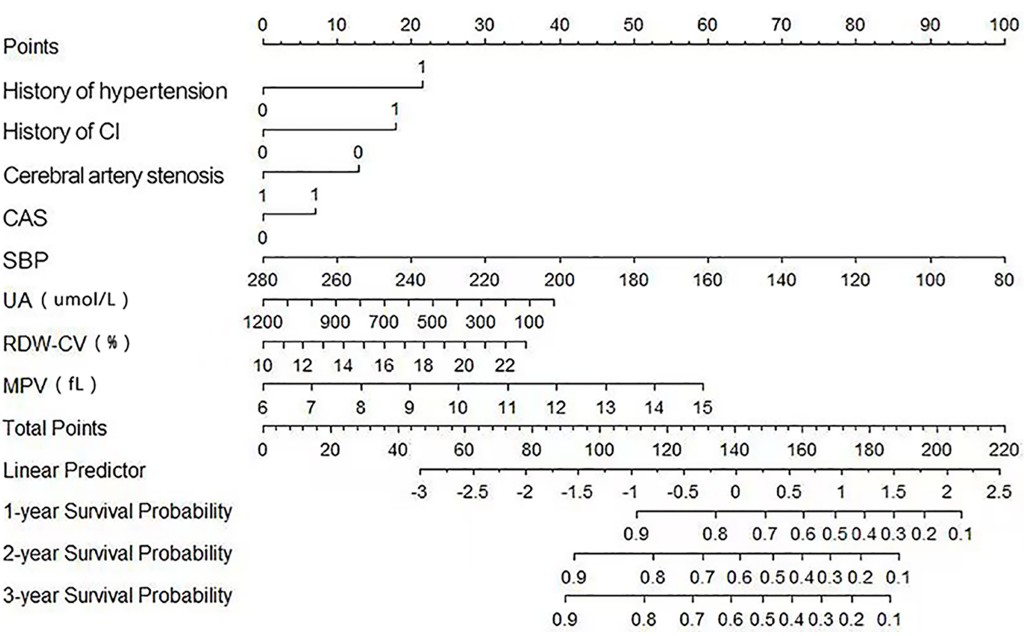

**Figure 2 Nomograms of predictive models for AIS recurrence risk.**

ascertained by the survival analysis random forest model, collectively included in a multivariate Cox regression model to establish a more comprehensive and precise AIS recurrence risk prediction model (Model 2). Compared to Model 1, Model 2 not only offers a more extensive and in-depth variable selection but also a more rigorous statistical approach, as it incorporates various advanced statistical techniques to optimize the variable selection process. Furthermore, Model 2, through multivariate Cox regression analysis, accounts for the interplay and synergistic effects among variables, which is challenging to capture in traditional univariate analyses. This multivariate analysis provides a more accurate risk assessment, aiding in the identification of significant risk factors that may be overlooked in univariate analyses. Additionally, we adopted survival analysis methods to estimate readmission rates at 1, 2, and 3 years, offering a temporal perspective on risk assessment and allowing for more dynamic and personalized intervention timing for clinicians. Consequently, Model 2 demonstrates significant advantages in the accuracy of risk prediction and the practicality of clinical application, providing new scientific evidence for the prevention and treatment of AIS recurrence.

Our predictive model has revealed that hypertension, history of cerebral infarction, cerebral artery stenosis, carotid atherosclerosis, SBP, RDW-CV, MPV, and UA are key factors in the recurrence of AIS. Under hypertensive conditions, inflammatory mediators such as tumor necrosis factor-alpha (TNF-α) and interleukin-6 (IL-6) released by vascular endothelial cells activate the NF-κB signaling pathway, promoting the development of vascular inflammation and atherosclerosis (*Chen et al., 2019*; *Ma et al., 2023b*; *Xu, Wang & Luo, 2021*). Additionally, the activation of the renin-angiotensin system (RAS) is closely related to blood pressure regulation and vascular remodeling, and its abnormal activation may lead to vascular stenosis and stroke recurrence (*Maïer et al., 2022*; *Roth et al., 2023*).

Furthermore, a history of cerebral infarction, cerebral artery stenosis, and carotid atherosclerosis can lead to a reduction in cerebral blood flow, exacerbating local cerebral ischemia and hypoxia. Local ischemia activates the hypoxia-inducible factor 1α (HIF-1α) pathway, which plays a key role in cellular adaptation to hypoxic environments, but its sustained activation is also associated with vascular remodeling and thrombosis (*Li, Tao & Wu, 2022*; *Ma et al., 2023a*). Genetic studies have also shown that polymorphisms related to vascular diseases may affect an individual's susceptibility to arterial stenosis (*Chen et al., 2018*; *Wakim et al., 2021*). An elevated RDW-CV, as an indicator of red blood cell size variability, usually indicates variability in red blood cell size, which may be related to iron metabolism disorders, affecting the maturation and function of red blood cells, and thus impacting the capacity for oxygen transport. Genetic variations in the hemochromatosis gene (HFE) are associated with abnormalities in iron metabolism (*Li, Zhou & Tang, 2017*), and these variations may indirectly affect the risk of AIS recurrence by impacting the maturation and function of red blood cells. An increase in MPV indicates the activation state of platelets, which involves multiple signaling pathways, including the interaction of platelet membrane glycoproteins. Overactivation of platelets can lead to the formation of microvascular thrombi (*Du Plooy et al., 2013*), thereby increasing the risk of stroke recurrence. Uric acid, as a metabolic product in the blood, has elevated levels associated with various cardiovascular diseases, and it can increase the risk of stroke by promoting endothelial dysfunction, increasing oxidative stress, and inflammatory responses.

We followed 523 AIS patients for 3 years and elicited that the recurrence rate was 15.1% (79/523) within 1 year, 6.1% (32/523) within 2 years, and 0.57% (3/523) within 3 years. This is in agreement with the study of *Lotlikar et al. (2022)*, which shows that the stroke recurrence rate was highest in 1 year after the initial onset and then decreased yearly. However, the reduction in recurrence rate over 3 years in our research was greater than the findings of *Strambo et al. (2021)*. This discrepancy may be due to differences in study samples, geographic location, treatment, or duration of follow-up. Nonetheless, our study further emphasizes the importance of ongoing management of stroke patients to reduce their recurrence risk. Our research further revealed that hypertension and carotid atherosclerosis were major risk factors for stroke recurrence. These findings suggest that our model can categorize patients into different risk levels according to risk factors, thus achieving more refined risk management. For patients identified as high-risk, it is essential to enforce rigorous blood pressure management and antiplatelet therapy. These interventions can significantly reduce the risk of AIS recurrence, thereby providing more effective protection for these patients. Furthermore, our findings advocate for incorporating a comprehensive assessment of carotid health status into the standard post-stroke evaluation framework. This assessment is vital for identifying patients who may benefit from early interventions. For example, carotid endarterectomy or stenting are pivotal early interventions to reduce the risk of stroke resulting from carotid stenosis. These interventions enable the development of personalized secondary prevention strategies for patients, thereby improving the quality of life for stroke survivors and reducing the recurrence rate of strokes. Our research not only provides valuable

information for clinicians but also introduces innovative approaches for the management of post-stroke patients.

There are also some limitations. First, variables included were relatively limited, which may impact the accuracy of the model. This limitation may stem from specific constraints in the data collection process, like incomplete physiological characteristics at admission and incomplete testing items, which may prevent us from fully capturing the multivariate characteristics of all stroke patients. Second, our COX model mainly captured the linear relationship between variables and outcomes and failed to consider the possible non-linear relationships. This is a limitation of the model itself but also underscores further exploration of these non-linear relationships in future studies using spline functions or artificial intelligence methods. Despite these limitations, our research methodology has some strengths. Visualizing the final variables screened through the nomograms can intuitively and simply interpret the model, which provides valuable direction for future research and benefits clinical practice. In future studies, we need to further increase the number and type of variables, including node-related data at the time of discharge from the first hospitalization as well as information during follow-up, to improve the accuracy of the model. Also, future research should explore more complex models to capture possible non-linear relationships between variables and outcomes. With these improvements, we hope to further enhance the predictive power of the model and provide stronger support for the treatment and rehabilitation of stroke patients.

## CONCLUSIONS

In summary, we used self-help sampling, LASSO regression, and random forest methods to construct a recurrence risk prediction model of acute ischemic stroke (AIS), which included eight variables including history of hypertension, history of cerebral infarction, cerebral artery stenosis, carotid atherosclerosis, systolic blood pressure, RDW-CV, MPV, and UA. In addition to personal and life characteristics, laboratory indicators such as RDW-CV, MPV, and UA are easily obtainable biomarkers in routine examinations, and highly suitable for clinical application in primary and community hospitals. Moreover, these models demonstrated good discriminative ability in the C-index, indicating their potential predictive capacity. We anticipate that these models will serve as effective tools for preventing recurrence in stroke patients and providing decision support for physicians.

## LIST OF ABBREVIATIONS

| | |
|---|---|
| **AIS** | Acute Ischemic Stroke |
| **CI** | Cerebral Infarction |
| **C-index** | concordance index |
| **M(IQR)** | Median of Interquartile Range |
| **LASSO** | Least Absolute Shrinkage and Selection Operator |
| **SBP** | Systolic Blood Pressure |
| **DBP** | Diastolic Blood Pressure |
| **NIHSS** | NIH Stroke Scale score |
| **WBC** | White blood cell count |

| | |
|---|---|
| **LYMPH%** | Lymphocyte percentage |
| **NEUT%** | Neutrophil percentage |
| **MONO%** | Monocyte percentage |
| **EO%** | Eosinophil percentage |
| **BASO%** | Basophil percentage |
| **RBC** | Red blood cell count |
| **HGB** | Hemoglobin |
| **HCT** | Hematocrit |
| **MCV** | Mean corpuscular volume |
| **MCH** | Mean corpuscular hemoglobin |
| **MCHC** | Mean corpuscular hemoglobin concentration |
| **RDW-CV** | Red blood cell distribution width-coefficient of variability |
| **RDW-SD** | RDW standard deviation |
| **PLT** | Platelet count |
| **PDW** | Platelet distribution width |
| **MPV** | Mean platelet volume |
| **P-LCR** | Platelet large cell ratio |
| **PCT** | Plateletcrit |
| **CRP** | Ultrasensitive C-reactive protein |
| **PT** | Prothrombin time |
| **Fg** | Fibrinogen |
| **TT** | Thrombin time |
| **APTT** | Activated partial thromboplastin time |
| **D-Dimer** | D-Dimer |
| **CREA** | Creatinine |
| **BUN** | Urea |
| **UA** | Uric acid |
| **CYC** | Cystatin C |
| **$\beta$2-MG** | $\beta$2-microglobulin |
| **ALT** | Alanine aminotransferase |
| **AST** | Aspartate aminotransferase |
| **CK** | Creatine kinase |
| **GLU** | Random glucose |
| **TC** | Total cholesterol |
| **TG** | Triglycerides |
| **HDL-C** | High density lipoprotein cholesterol |
| **LDL-C** | Low density lipoprotein cholesterol |
| **HCY** | Homocysteine |
| **HbA1c** | Glycated Hemoglobin |
| **NLR** | Neutrophils/lymphocytes |
| **PLR** | Platelets/lymphocytes |

| MHR | Monocytes/HDL |
| CAS | Carotid Artery Stenosis |

### Funding

This work was supported by the Research project of Guangxi Zhuang Autonomous Region Health Commission (No. Z20210530). The funders had no role in study design, data collection and analysis, decision to publish, or preparation of the manuscript.

### Grant Disclosures

The following grant information was disclosed by the authors:
Research Project of Guangxi Zhuang Autonomous Region Health Commission: Z20210530.

### Competing Interests

The authors declare that they have no competing interests.

### Author Contributions

- Liuhua Ke conceived and designed the experiments, performed the experiments, analyzed the data, prepared figures and/or tables, and approved the final draft.
- Hongyu Zhang conceived and designed the experiments, performed the experiments, prepared figures and/or tables, and approved the final draft.
- Kang Long performed the experiments, prepared figures and/or tables, and approved the final draft.
- Zheng Peng conceived and designed the experiments, analyzed the data, authored or reviewed drafts of the article, and approved the final draft.
- Yongjun Huang analyzed the data, authored or reviewed drafts of the article, and approved the final draft.
- Xingxuan Ma analyzed the data, authored or reviewed drafts of the article, and approved the final draft.
- Wanjun Wu performed the experiments, prepared figures and/or tables, and approved the final draft.

### Human Ethics

The following information was supplied relating to ethical approvals (*i.e.*, approving body and any reference numbers):
    Liuzhou Hospital of Traditional Chinese Medicine.

### Data Availability

    The raw data is available in the Supplemental File.

## Clinical Trial Registration

The following information was supplied regarding Clinical Trial registration:
2021MAY-KY-YN-009-01.

## Supplemental Information

Supplemental information for this article can be found online at http://dx.doi.org/10.7717/peerj.18605#supplemental-information.

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
