# Peer review of "Risk factors and prediction models for recurrent acute ischemic stroke: a retrospective analysis"

_PeerJ, doi:10.7717/peerj.18605_

## Round 0.1 · original submission · Major Revisions

As you can see, both reviews raised critical points that have to be correct. I highlight firstly the necessity of including additional studies germane to yours; the question about the novelty of your study has to be considered, inclusion of additional data seems also to be critical.

Reviewer 1 ·

Basic reporting

As the reviewer for your manuscript titled "Risk factors for recurrence of acute ischemic stroke and development of risk prediction models", I would like to acknowledge the significant contribution your research makes to the field. Following a comprehensive evaluation of your submission, I have identified several areas requiring revision before the manuscript can be considered for final acceptance.

Experimental design

Major concerns:
1) The risk factor for recurrence of acute ischemic stroke and risk prediction model has been extensively explored in existing research, including machine learning, imaging, biomarkers, and etc. Consequently, this topic appears to exhibit limited novelty.
Wang K, Shi Q, Sun C, Liu W, Yau V, Xu C, Liu H, Sun C, Yin C, Wei X, Li W, Rong L. A machine learning model for visualization and dynamic clinical prediction of stroke recurrence in acute ischemic stroke patients: A real-world retrospective study. Front Neurosci. 2023 Mar 27;17:1130831. doi: 10.3389/fnins.2023.1130831
Abedi V, Avula V, Chaudhary D, Shahjouei S, Khan A, Griessenauer CJ, Li J, Zand R. Prediction of Long-Term Stroke Recurrence Using Machine Learning Models. J Clin Med. 2021 Mar 20;10(6):1286. doi: 10.3390/jcm10061286
Elhefnawy ME, Sheikh Ghadzi SM, Albitar O, Tangiisuran B, Zainal H, Looi I, Sidek NN, Aziz ZA, Harun SN. Predictive model of recurrent ischemic stroke: model development from real-world data. Front Neurol. 2023 Apr 28;14:1118711. doi: 10.3389/fneur.2023.1118711
Liu J, Wu Y, Jia W, Han M, Chen Y, Li J, Wu B, Yin S, Zhang X, Chen J, Yu P, Luo H, Tu J, Zhou F, Cheng X, Yi Y. Prediction of recurrence of ischemic stroke within 1 year of discharge based on machine learning MRI radiomics. Front Neurosci. 2023 May 4;17:1110579. doi: 10.3389/fnins.2023.1110579

2) In the study, this article also mentioned that previous studies have certain limitations (line 70-74). Please provide a brief example to illustrate the deficiencies of previous studies, as well as the advantages and innovations of this study compared to previous studies.

3) Although there are many variables included in this study, some crucial indicators are still lacking, such as echocardiography, specific indicators of vascular ultrasound (including plaque stability/instability, plaque diameter, plaque location, stenosis degree, blood flow velocity, etc.), cranial MRI, cranial MRA, pre admission emergency electrocardiogram, and some key laboratory indicators such as creatinine clearance (eGFR), BNP, etc. The lack of these important indicators may make the prediction model lose greater persuasiveness.

4) In line 126-131 of statistical analyses section, this article mentioned “when the percentage of missing variables reached 80% or above, the variables were directly deleted”. Will this lead to a lack of credibility in the research results, and what is the basis for handling missing variables in this way? Please provide specific explanations.

5) I have found a serious error in the table of this article. The data presented in Table 1 may be incorrect. Please carefully check the data in Table 1, as it is inconsistent with the results description, such as history of hypertension, history of CI, SBP, DBP, NIHSS score and etc. Please correct the error timely and explain the reasons.

Validity of the findings

Minor points:
1) In this article, it does not elaborate on the detailed process of variable screening. a detailed explanation should be provided in the section of materials & methods, preferably accompanied by a screening flowchart.

2) If the follow-up period and content (specific indicators for follow-up) should be explained in detail.

3) It would be better to expand the sample size, especially in the absence of an external validation cohort. External validation is the use of unused data in model development to evaluate the performance of the model in new data. Compared to internal validation, external validation focuses more on the model's transportability and generalizability.

·

Basic reporting

• Clarity and Language: The manuscript is generally well-written in clear, professional English, though minor language edits are recommended for fluency and readability.
• Introduction: It could benefit from further elaboration on how the proposed prediction models add value beyond existing models. A clearer articulation of the study's novelty and rationale is suggested.
• Literature Review: Some references could be updated or expanded to include more recent findings related to AIS recurrence risk factors and prediction models.

Experimental design

• The study falls within the journal's scope and addresses a significant research question regarding AIS recurrence. The experimental design involves statistical methods such as LASSO regression and machine learning. However, the description of the machine learning models and validation processes could be more detailed to allow replication by other researchers. For example, details on model hyperparameters, training-test split ratios, and any cross-validation techniques used would be helpful.

Validity of the findings

• Data and Statistical Analysis: The statistical analysis is generally robust, with appropriate use of multivariate Cox regression and LASSO regression methods to identify significant risk factors. The manuscript demonstrates that the predictive models (Model 1 and Model 2) are statistically sound and controlled. However, the authors should address the potential for overfitting in Model 2 due to the inclusion of numerous variables.
• Model Validation: Consider including an external validation cohort to assess the generalizability of the predictive models.
• Results Interpretation: The interpretation of the findings is reasonable and aligns with the data presented. The manuscript successfully identifies key risk factors (e.g., hypertension, cerebral infarction history, and carotid atherosclerosis) and explains their clinical significance. Nevertheless, the discussion could benefit from further elaboration on the clinical implications of these findings and how the models could be integrated into clinical practice.

---

## Round 0.2 · accepted · Accept

I confirm that the article is now Acceptable.

Note that the reviewer has indicated a small error in the abstract.

·

Basic reporting

The authors have addressed all my questions.
Minor: There is an extra "background" in the abstract.

Experimental design

No comment

Validity of the findings

No comment

Additional comments

No comment